# HORSE: Hierarchical Representation for Large-Scale Neural Subset Selection

**Binghui Xie, Yixuan Wang, Yongqiang Chen, Kaiwen Zhou, Yu Li, Wei Meng, James Cheng**
Department of Computer Science and Engineering
The Chinese University of Hong Kong

## Abstract

Subset selection tasks, such as anomaly detection and compound selection in AI-assisted drug discovery, are crucial for a wide range of applications. Learning subset-valued functions with neural networks has achieved great success by incorporating permutation invariance symmetry into the architecture. However, existing neural set architectures often struggle to either capture comprehensive information from the superset or address complex interactions within the input. Additionally, they often fail to perform in scenarios where superset sizes surpass available memory capacity. To address these challenges, we introduce the novel concept of the *Identity Property*, which requires models to integrate information from the originating set, resulting in the development of neural networks that excel at performing effective subset selection from large supersets. Moreover, we present the Hierarchical Representation of Neural Subset Selection (HORSE), an attention-based method that learns complex interactions and retains information from both the input set and the optimal subset supervision signal. Specifically, HORSE enables the partitioning of the input ground set into manageable chunks that can be processed independently and then aggregated, ensuring consistent outcomes across different partitions. Through extensive experimentation, we demonstrate that HORSE significantly enhances neural subset selection performance by capturing more complex information and surpasses the state-of-the-art methods in handling large-scale inputs by a margin of up to 20%.

## 1 Introduction

Set-valued functions are of great importance to a wide range of real-world applications. For example, anomaly detection aims to identify a set of outliers from a larger dataset that could be users or financial transactions [Zhang et al., 2020]. Another example is the recommender system, where the objective is to identify a set of products that better satisfy customer preferences [Ou et al., 2022]. In these scenarios, there is a need for implicitly learning a set function [Rezatofighi et al., 2017, Zaheer et al., 2017] that quantifies the usefulness of a given subset of the inputs. The set function assigns a utility value to each subset, and the subset with the highest utility corresponds to the most desired output.

To illustrate the concept, let us consider the task of a recommender system. In this task, we aim to recommend a subset of items $S$ from a larger item pool $V$, denoted as $S \in V$, that maximizes the utility of $S$ with respect to the satisfaction of the customers. The utility can be captured by a parameterized utility function, denoted as $F_\theta(S; V)$, and our goal is to optimize the following criteria:

$$S^* = \arg\max_{S \in 2^V} F_\theta(S; V). \tag{1}$$

One straightforward method [Balcan and Harvey, 2018] involves explicitly modeling the utility by learning the function $U = F_\theta(S; V)$ using supervised data. This data consists of pairs

38th Conference on Neural Information Processing Systems (NeurIPS 2024).

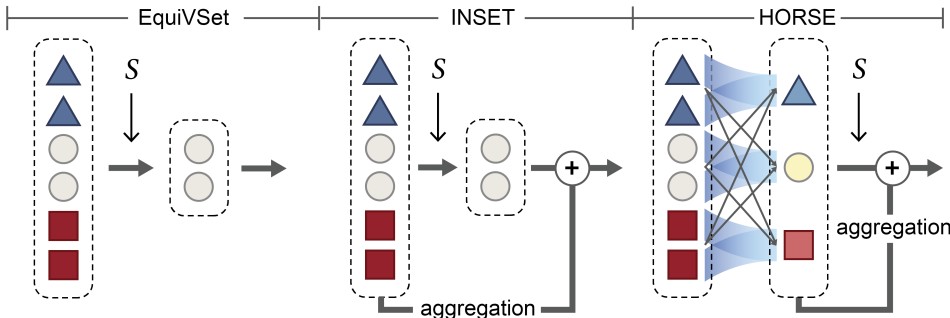

Figure 1: Comparison of HORSE to the state-of-the-arts EquiVSet and INSET in handling subsets. "S" represents the supervision, indicating the specific subset of interest. "+" refers to the aggregation of different vectors, which is implemented through concatenation in practice. Unlike EquiVSet and INSET, the HORSE model captures more complex information from $V$ by employing attention mechanisms. Furthermore, HORSE facilitates the division of $V$ into distinct partitions.

$\{(S_i, V_i), U_i\}_{i=1}^N$, where $U_i$ represents the actual utility value of subset $S_i$ given the respective item pool $V_i$. However, this training approach becomes challenging to implement due to the significant number of supervision signals required, which can be expensive and time-consuming to acquire [Ou et al., 2022]. To overcome this limitation, an alternative approach is to tackle Eq. 1 using an implicit learning method from a probabilistic perspective [Tschiatschek et al., 2018]. This approach requires utilizing data in the form of $\{(V_i, S_i^*)\}_{i=1}^N$, where $S_i^*$ represents the optimal subset corresponding to $V_i$. The objective is to estimate the parameters $\theta$ such that Eq. 1 holds for all possible $(V_i, S_i)$. In practical training, with limited data sampled from the underlying distribution $P(S, V)$, the empirical log-likelihood $\sum_{i=1}^N [\log p_\theta(S^*|V)]$ is maximized for all the data pairs $\{S, V\}$, where $p_\theta(S|V)$ is proportional to $F_\theta(S; V)$ for all $S \in 2^V$ [Ou et al., 2022, Xie et al., 2024]. Additional details are available in Appendix D.3.

The crux of the matter lies in determining the structure of neural networks that can effectively model $F_\theta(S, V)$ throughout the entire process. One commonly employed approach in the literature is to utilize an encoder to generate feature vectors for each element in $V$. These vectors are then inputted into DeepSets [Zaheer et al., 2017], along with the corresponding supervised subset $S$, in order to learn the permutation invariant set function $F(S)$. However, this methodology may neglect the interaction between $S$ and $V$, leading to a reduction in the expressive capacity of the models. In the previous study, Xie et al. [2024] suggest incorporating the sum-pooling representation of $V$ into $S$ to enhance the performance. Yet, the simple integration in Xie et al. [2024], Wang et al. [2024b] limits its capacity to effectively model interactions among elements or subsets within these sets. Furthermore, the approach struggles with high-cardinality sets $V$, as encoding the entire set into memory may not be feasible [Bruno et al., 2021].

To address these problems, and inspired by [Willette et al., 2023], we introduce the notion of the *Identity Property*, a desirable concept for the effective functioning of the model $F(S, V)$. Identity Property requires $F(S, V)$ to accurately reflect which set $V$ the information $S$ originates from. In order to capture the interplay between $S$ and $V$ by adhering to the Identity Property, we propose a subset-based attentive set encoder. Additionally, this encoder facilitates the division of a large set $V$ into smaller and manageable subsets. These subsets can be processed independently and later aggregated, ensuring no loss of information from $V$. Hence, our approach is able to efficiently handle large-scale subset representation learning. As depicted in Figure 1, our method is capable of modeling more complex information and managing large-scale inputs more effectively than the two state-of-the-art approaches in the field of Neural Subset Selection tasks, EquiVSet [Ou et al., 2022] and INSET [Xie et al., 2024].

In this work, we make several contributions to the field of neural subset selection, which can be summarized as follows:

- We introduce and rigorously define a critical concept termed as the *Identity Property* for neural subset selection. This property requires that models can reliably determine the source

set $V$ from which the information of the subset $S$ is derived, which is a crucial requirement for neural subset selection tasks.

- To adhere to the Identity Property and model complex interaction, we present a subset-based attention mechanism. This mechanism is crafted to learn the Hierarchical Representation of Neural Subset Selection, denoted as HORSE. Our theoretical analysis confirms that HORSE not only upholds the Identity Property but also maintains Permutation Invariance.

- Through extensive empirical research, we validate the effectiveness of HORSE. Our experiments across a variety of datasets demonstrate the consistently superior performance of HORSE. Additionally, we specifically explore HORSE's capabilities in large set environments, further showcasing its practical applicability and efficiency compared with the baselines.

## 2 Related Work

### 2.1 Set Encoding.

The exploration of network architectures tailored for set-structured inputs has become a vibrant area of research in recent years. A number of key studies [Ravanbakhsh et al., 2017, Edwards and Storkey, 2017, Zaheer et al., 2017, Qi et al., 2017, Horn et al., 2020, Bloem-Reddy and Teh, 2020, Wang et al., 2023] have laid the groundwork in this domain, primarily focusing on creating models that are permutation equivariant using conventional feed-forward neural networks. These foundational models have been successful in universally approximating continuous permutation-invariant functions, primarily through the application of set-pooling layers to aggregate information across different elements of a set regardless of their order.

However, these methodologies have primarily concentrated on learning representations at the aggregate set level, paying less attention to more nuanced interactions occurring at elements. Recognizing this gap, more recent research efforts have aimed at introducing more sophisticated interaction modeling within invariant set functions for various applications. A notable example is the work by [Lee et al., 2019b], which incorporates self-attention mechanisms to facilitate the processing of elements within sets, thereby effectively capturing element-wise interactions. Moreover, the concept of Janossy pooling, proposed by [Murphy et al., 2018], introduced a novel approach to incorporate higher-order interactions within the pooling process. Since then, subsequent studies have built upon this advancement, leading to further refinements and innovations in the field, e.g., [Kim, 2021, Li et al., 2020, Bruno et al., 2021, Willette et al., 2023].

### 2.2 Hierarchical Set Function.

The existing literature primarily concentrates on processing entire input sets, often overlooking the information provided by the sub-levels. Addressing this oversight, Maron et al. [2020] introduced an innovative approach that integrates the symmetry of elements to generate representations of an input set. This methodology was further expanded into a broader context by Wang et al. [2020]. Moreover, Bevilacqua et al. [2022] proposed a novel framework aimed at enhancing graph representations by including whole-graph representations to encode each subgraph. Along similar lines, Xie et al. [2024] developed an information aggregation module designed to learn $F(S, V)$ effectively.

Despite these advancements, a gap remains in the current research landscape. These methods tend to overlook more complex interactions between elements or subsets within sets. Furthermore, they often fall short in scenarios where the input set has a significantly large cardinality, indicating a need for more scalable and interaction-sensitive approaches in set processing. In Table 1, we compare our proposed method with importance baselines commonly used in subset selection tasks. Specifically, DeepSets

Table 1: Properties of Various Methods: "Attn" indicates the use of the attention mechanism, "V" signifies the explicit utilization of information from $V$, and 'Large-scale' denotes the capability of the methods to generalize effortlessly to large-scale settings.

| Model | Attn | V | Large-Scale |
|---|---|---|---|
| DeepSets [Zaheer et al., 2017] | ✗ | ✗ | ✓ |
| Set Transformer [Lee et al., 2019a] | ✓ | ✓ | ✗ |
| EquiVSet [Ou et al., 2022] | ✗ | ✗ | ✓ |
| INSET [Xie et al., 2024] | ✗ | ✓ | ✗ |
| HORSE | ✓ | ✓ | ✓ |

can handle large-scale settings but may lose complex information due to its simple pooling-based structure. Set Transformer excels at modeling complex information within sets but faces challenges with large input set sizes. Methods tailored for subset selection tasks, like EquiVSet and INSET, may struggle with learning intricate interactions and often overlook large-scale settings. HORSE is designed to address these drawbacks.

## 2.3 Core Subset Selection

Recent work has focused on extracting subsets from training datasets to decrease cost and improve effectiveness [Wei et al., 2015, Mirzasoleiman et al., 2020, Yang et al., 2023]. This research also highlights the importance of modeling the relationship between the original dataset and its subsets. Unlike neural subset selection, these core subsets are unlabeled, and typically, more data in the core subset enhances the performance of the models. Our approach differs in that our optimal subset is labeled within the training set, and its size is constrained.

# 3 Method

## 3.1 Preliminaries

In this paper, we concentrate on the development of neural networks for the purpose of modeling the hierarchical set function $F(S, V)$, a critical component for tasks involving sets, such as neural subset selection. For every ground set $V$, assumed to consist of $n$ elements represented as $x_i$, that is, $V = \{x_1, x_2, ..., x_n\}$, each element $x_i$ belonging to $\mathcal{X}$ is characterized by a $d$-dimensional tensor. Typically, the ground set $V$ can be conceptualized as an assembly of multiple disjoint subsets, explicitly $V = S_1 \cup S_2 \cup \cdots \cup S_m$, where $S_i \cap S_j = \emptyset$ for $i \neq j$ and each $S_i$ is a subset in $\mathbb{R}^{n_i \times d}$. In this context, $n_i$ denotes the number of elements in subset $S_i$. Generally, $S \subseteq V$ acts as a supervisory signal in the form of a mask over $V$ to indicate the elements to be selected. For the sake of clarity, we define $S$ as the concrete subset derived from this mask.

In the context of neural subset selection, the task entails the encoding of subsets $S_i$ into representative vectors to forecast the associated function value $Y \in \mathcal{Y}$. Traditional approaches, such as those documented in Zaheer et al. [2017] and Ou et al. [2022], involve directly selecting $S_i$ based on the encoding embeddings of all elements within $V$, subsequently feeding $S_i$ into feed-forward networks. Nonetheless, these methods model the function $F(S_i, V)$ solely based on the explicit subsets $S_i$, potentially leading to suboptimal results due to the omission of the broader context provided by the ground set $V$. This section introduces a novel attention-based method for encoding subset representations, which distinctively incorporates information from the entire input set $V$, thereby enhancing performance.

## 3.2 Identity Property

To effectively model $F(S, V)$, Xie et al. [2024] have proposed to combine the representations of $V$ and $S$. In practice, this involves utilizing two DeepSets architectures, as proposed by Zaheer et al. [2017], to independently process $S$ and $V$ before merging their outputs, as presented by Figure 1. Given that set pooling operations process each element independently, certain information about the interactions among elements is inevitably lost. This omission can render some problems more challenging than necessary. To address this issue and facilitate the learning of complex interactions within sets, we introduce the following principles:

**Property 3.1.** *Consider $V \in \mathbb{R}^{n \times d}$ and $S \subseteq V$ where $S \in \mathbb{R}^{s \times d}$, assuming that $V$ is partitioned into a random collection of disjoint subsets $V = S_1 \cup S_2 \cup \cdots \cup S_m$. Here, $m$ varies within the range $[1, n]$, dependent on the chosen method of partitioning. The function $F$ is said to satisfy the Identity Property if and only if there exist functions $g$ and $h$ such that*

$$F(h(S), h(V)) = F(h(S), g(h(S_1), \ldots, h(S_m))), \qquad (2)$$

*where $g$ serves as an aggregation function that effectively combines the encoded representations of the subsets, ensuring that $F$ leverages both the specific subset $S$ and the ground set $V$ through the transformations applied by $h$ and the aggregation by $g$.*

The method introduced by Xie et al. [2024] is notable for satisfying the Identity Property through its utilization of sum-pooling to simultaneously process all elements. However, this approach may

not be practical for scenarios involving large inputs and may struggle to capture more complex information. In response to these limitations, we propose an attention-based method designed to fulfill the requirements of the Identity Property. Additionally, our interpretation of this property accommodates scenarios where $S$ differs from the union of subsets $\{S_1 \cup S_2 \cup \cdots \cup S_m\}$. In practice, $S$ is often chosen to be $S_1$ for simplicity. This nuanced approach allows for greater flexibility and effectiveness in encoding set information, especially in complex or large-scale settings.

### 3.3 Attention-Based Set Representation

In this section, we introduce a formulation for an attention-based set encoding function $F$, leveraging the concept of partitions (referred to as slots in [Bruno et al., 2021, Willette et al., 2023]). Given a ground set $V \in \mathbb{R}^{n \times d}$, we randomly divide it into $m$ subsets. For each subset, we allocate a unique embedding $s_i \in \mathbb{R}^{d_s}$. Furthermore, we establish $\zeta = [s_1, \ldots, s_m]^T$ as a matrix in $\mathbb{R}^{m \times d_s}$. Similar to [Willette et al., 2023], we initialize $\zeta$ by sampling $m$ embeddings $s_i$ from a parameterized Gaussian distribution with random initialization. Following this setup, we calculate the unnormalized attention scores between $\zeta$ and $V$, facilitating a dynamic weighting of elements within $V$ based on their relevance to each partition's embedding. This process aims to capture the nuanced interrelations within subsets and between elements and their corresponding subsets.

$$q = LN(\zeta W^q), \tag{3}$$
$$k = VW^k,$$
$$v = VW^v, \tag{4}$$

In this expression, "$LN$" represents Layer Normalization, and the matrices $W^q \in \mathbb{R}^{d_s \times d_h}$, $W^k \in \mathbb{R}^{d \times d_h}$, and $W^v \in \mathbb{R}^{d \times d_h}$ are introduced. These matrices serve to project $V$ and $\zeta$ into a shared dimensional space $d_h$. Subsequently, we employ a dot product attention mechanism to assess the interactions between $V$ and $\zeta$. This process is governed by the specified formulation, strategically aligning elements of $V$ with the embeddings in $\zeta$ through dimensional congruence, thus enabling a nuanced, attention-driven analysis of set elements in relation to their partitioned subsets.

$$\hat{M} = \sqrt{d_h^{-1}} \cdot qk^T, \tag{5}$$
$$\hat{A} = \sigma(\hat{M}) \in \mathbb{R}^{m \times n}, \tag{6}$$

where $\sigma$ denotes an element-wise activation function. Utilizing the unnormalized attention scores, denoted by $\hat{A}$, we proceed to define the following mapping operation:

$$\bar{h} = \text{nl}(\hat{A})v. \tag{7}$$

In this context, $\bar{h}$ signifies a transformation function mapping from $\mathbb{R}^{n \times d}$ to $\mathbb{R}^{m \times d_h}$. The term "nl" represents a normalization operation, defined as follows:

$$\text{nl}(\hat{A})_{i,j} = \hat{A}_{i,j} / \sum_{i=1}^{m} \hat{A}_{i,j}, \tag{8}$$

which normalizes the column of $\hat{A}$. Then, we can apply a pooling function (such as sum, mean, min, or max) across the columns of $\bar{h}(V)$ and select the sigmoid function for $\sigma$, thereby establishing an attention mechanism akin to the SSE (Set Stream Embedding) method proposed by Bruno et al. [2021]. However, this approach has its limitations. Given that the attention score $\text{nl}(\hat{A})_{i,j}$ is calculated independently of the other $n-1$ attention scores, it is not feasible for the rows of $\text{nl}(\hat{A})$ to form convex coefficients, unlike the softmax outputs typically observed in conventional attention mechanisms, as described by Willette et al. [2023].

To address this issue, we follow Willette et al. [2023] to aggregate information across all rows of $\text{nl}(\hat{A})$, thereby incorporating dependencies among different set elements into the attention mechanism. This is achieved through a specific normalization process, outlined as follows:

$$M = diag(\text{nl}(\hat{A})\mathbf{1}_n)^{-1} \tag{9}$$

where $\mathbf{1}_n = (1, \ldots, 1) \in \mathbb{R}^n$ represents a vector of ones of dimension $n$, and $M_2 \in \mathbb{R}^m$ signifies a vector in an $m$-dimensional space. Subsequently, we can compute $h(V)$ by applying the normalization

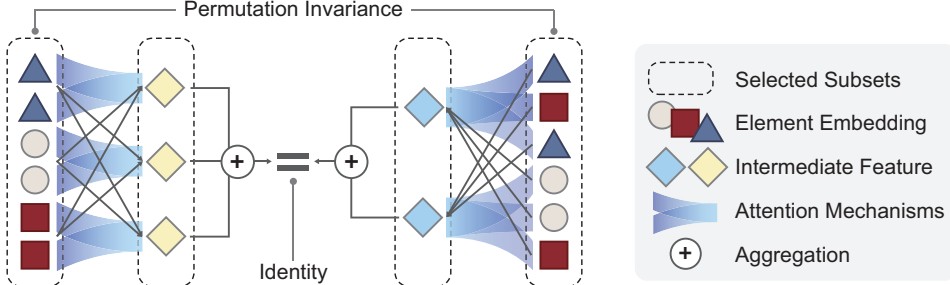

Figure 2: This figure illustrates the HORSE model's capability to achieve Permutation Invariance and satisfy the Identity Property in subset selection tasks. It demonstrates that HORSE maintains consistent output despite the permutation of input set elements and the partition if the ground set.

term $M$, as follows:

$$h(V) = M\mathrm{nl}(\hat{A})VW^v. \tag{10}$$

Since $S_i$ is a subset of $V$ based on a partition method, $h_V(S_i)$ can be derived from $h(V)$. Detailed steps are provided in Algorithm 1. For simplicity, we omit $V$ and use $h(S_i)$ going forward. This process ensures that $h$ meets the criteria specified in Eq. 2. By constructing such an $h$ function, we ensure that the model can recognize the input set $V$ regardless of its partitioning, leading to the property that $g(h(S_1), h(S_2), \ldots, h(S_m))$ yields the same value for any partition of $V$.

Furthermore, it facilitates the learning of interactions among the partitioned segments of $V$, essentially enabling the model to identify the characteristics of the input ground set $V$. Specifically, we concatenate $h(S)$ with $g(h(S_1), h(S_2), \ldots, h(S_m))$. In practice, a Multilayer Perceptron (MLP) is utilized to process the concatenated tensor into a vector $Z \in \mathbb{R}^{d_o}$, achieving the following:

$$Z = F(h(S), g(h(S_1), h(S_2), \ldots h(S_m))) \in \mathbb{R}^{d_o}. \tag{11}$$

Given that the aforementioned process delineates the entire calculation in a matrix format, which may be complex for some readers, we have taken steps to enhance comprehension. To better illustrate how our method establishes an attention map between subsets $S_i$, we have detailed the procedural steps in the Appendix (see Algorithm 1), with a particular focus on the generation of $h(S_i)$. This will not only clarify the operational details but also emphasize the underlying methodology and thought process involved in constructing $h(S_i)$.

## 4 Theoretical Results

In the realm of machine learning, particularly within the scope of set-based tasks, a fundamental requirement is the invariance to the permutation of input set elements. This characteristic ensures that the computation or outcome of a task is unaffected by the order in which the set's elements are presented, a principle that is especially pertinent to neural subset selection tasks. To address and formalize this aspect within the context of our proposed method, we present a theorem that rigorously demonstrates the permutation invariance of our approach.

**Theorem 4.1.** *Let $\mathbb{S}_n$ denote the set of all permutations of a given set $V$. Since $V \in \mathbb{R}^{n \times d}$ is represented by a matrix, let $\pi_V \in \mathbb{R}^{n \times n}$ be a random permutation applied to $V$. Given that $S \subseteq V$ represents a subset of $V$, the permutation $\pi_V$ naturally induces a corresponding permutation $\pi_S$ on $S$. Under these conditions, HORSE exhibits permutation invariance, which is defined as:*

$$F(h(S), h(V)) = F(h(\pi_S \cdot S), h(\pi_V \cdot V)) \tag{12}$$

Theorem 4.1 assures that irrespective of how the elements in the input set $V$ are ordered, the output generated by HORSE remains consistent. This property is crucial for ensuring the reliability and applicability of our method across a wide spectrum of set-based tasks, where the inherent order of data points should not influence the task outcome. Furthermore, an illustration of the underlying concept is provided in Figure 2.

Table 2: Product recommendation results for 12 different product categories. The best results are indicated in bold black, while the second-best results are highlighted in blue. Due to space limitation, we use Set-T to denote Set Transformer.

| Categories | Random | PGM | DeepSet | Set-T | EquiVSet | INSET | HORSE |
|---|---|---|---|---|---|---|---|
| Gear | 7.7 | 47.1±0.4 | 37.9±0.5 | 64.7±0.6 | 72.5±1.1 | 80.8±1.2 | **83.2±1.3** |
| Bath | 7.6 | 56.4±0.8 | 41.8±0.7 | 71.6±0.5 | 76.4±2.0 | 86.2±0.5 | **87.6±1.0** |
| Toys | 8.3 | 4.41±0.4 | 42.1±0.5 | 62.5±2.0 | 68.4±0.4 | 76.9±0.5 | **77.4±0.9** |
| Media | 9.4 | 44.1±0.9 | 42.6±0.4 | 53.0±2.0 | 55.4±0.5 | 62.0±2.3 | **65.2±1.5** |
| Safety | 6.5 | 25.0±0.6 | 22.1±0.4 | 23.4±0.9 | 23.1±2.0 | 23.8±1.5 | **26.9±1.2** |
| Diaper | 8.4 | 58.3±0.9 | 45.1±0.3 | 78.9±0.5 | 82.8±0.7 | **88.3±0.7** | 88.0±0.8 |
| Health | 7.6 | 44.9±0.2 | 45.2±0.1 | 69.2±1.2 | 70.5±0.9 | 81.2±0.5 | **81.6±0.6** |
| Carseats | 6.6 | 23.1±1.0 | 21.2±0.8 | 22.0±1.0 | 22.3±1.9 | 23.0±2.4 | **24.8±2.2** |
| Bedding | 7.9 | 48.5±0.6 | 48.1±0.2 | 76.2±2.2 | 76.2±0.5 | 85.7±1.1 | **87.1±0.7** |
| Feeding | 9.3 | 56.3±0.8 | 42.8±0.2 | 75.3±0.6 | 81.9±0.9 | 88.5±0.5 | **90.3±1.1** |
| Apparel | 9.0 | 53.3±0.5 | 50.8±0.4 | 68.0±2.0 | 76.4±0.5 | 83.7±0.3 | **85.4±0.6** |
| Furniture | 6.5 | 17.5±0.7 | 16.8±0.2 | 17.6±0.8 | 16.2±2.0 | 16.7±3.5 | **18.1±1.5** |

**Theorem 4.2.** *If $\sigma$ represents a strictly positive element-wise activation function, then HORSE satisfies Property 3.1.*

By satisfying this property, HORSE ensures that its encoding captures both the individual characteristics of subsets within $V$ and the overarching structure of the entire set, thereby facilitating a more nuanced and comprehensive understanding of set information. The proofs is inspired by [Willette et al., 2023] and can be found in Appendix A.

## 5  Experiments

In this section, we aim to demonstrate that HORSE significantly outperforms baseline models in a suite of benchmarks tailored to Neural Subset Selection tasks. Subsequently, we extend our investigation to scenarios involving large-scale input settings. Due to the page limitation, we have included additional experiments in Appendix D.

**Evaluations.** To assess the performance of various methods, we employ the mean Jaccard coefficient (MJC) as the evaluation metric. This metric quantifies the similarity between the predicted subset $S'$ and the true subset $S^*$ for each data sample $(S^*, V)$. The Jaccard coefficient is calculated as follows: $JC(S^*, S') = \frac{|S^* \cap S'|}{|S^* \cup S'|}$, where the intersection and union operations determine the size of the overlap and the total unique elements in both sets, respectively. The MJC is then derived by averaging the Jaccard coefficient across all test set samples. Please note that all the following reported performance metrics are presented in percentages, with a default multiplication factor of $100\%$.

**Baselines.** We conducted experiments compared with several approaches: Random, PGM [Tschiatschek et al., 2018], DeepSet [Zaheer et al., 2017], Set Transformer [Lee et al., 2019a], EquiVSet [Ou et al., 2022], and IN-SET [Xie et al., 2024]. The Random approach represents the expected performance of a random guess. DeepSet and Set Transformer are well-known methods or frameworks that satisfy permutation invariance, making them suitable for Neural Subset Selection tasks. PGM, EquiVSet and INSET are specifically designed for subset selection tasks. More comprehensive details available in Appendix D.

Table 3: Performance results on the Two-Moons and Gaussian-Mixture datasets. Bolded numbers denote the best performance on each dataset

| Method | Two Moons | Gaussian Mixture |
|---|---|---|
| Random | 5.5 | 5.5 |
| PGM | 36.0 ± 2.0 | 43.8 ± 0.9 |
| DeepSet | 47.2 ± 0.3 | 44.6 ± 0.2 |
| Set Transformer | 57.4 ± 0.2 | 90.5 ± 0.2 |
| EquiVSet | 58.5 ± 0.3 | 90.7 ± 0.2 |
| INSET | 58.2 ± 0.3 | 90.9 ± 0.2 |
| HORSE | **60.2 ± 0.5** | **91.8 ± 0.2** |

Table 4: Empircal results of compound selection Tasks. Bolded numbers denote the best performance on each dataset. Due to space limitations, we use "Set-T" to denote Set Transformer.

| | Random | PGM | DeepSet | Set-T | EquiVSet | INSET | HORSE |
|---|---|---|---|---|---|---|---|
| PDBBind | 9.9 | 91.0±1.0 | 90.1±1.1 | 91.9±1.5 | 92.4±1.1 | 93.5±0.8 | **94.1 ± 0.7** |
| BindingDB | 9.0 | 69.0±2.0 | 71.0±2.0 | 71.5±1.0 | 72.1±0.9 | 73.4±1.0 | **74.2 ± 1.1** |
| PDBBind-2 | 7.3 | 35.0±0.9 | 32.3±0.4 | 35.5±1.0 | 35.7±0.5 | 37.1±1.0 | **43.2 ± 0.6** |
| BindingDB-2 | 2.7 | 17.6±0.6 | 16.5±0.5 | 18.3±0.4 | 18.8±0.6 | 19.8±0.5 | **21.3 ± 0.5** |
| Average | 7.23 | 53.15 | 52.48 | 54.30 | 54.75 | 55.95 | **58.20** |

## 5.1 Synthetic Experiments

Firstly, We validate the effectiveness of our models through experimental trials focused on learning set functions, using two synthetic datasets: the two-moons dataset [Pedregosa et al., 2011] with an added noise variance of $\sigma^2 = 0.1$, and and a Gaussian mixture represented as $\frac{1}{2}\mathcal{N}(\mu_0, \Sigma) + \frac{1}{2}\mathcal{N}(\mu_1, \Sigma)$.

Take the Gaussian mixture as an example, the data generation procedure as follows: i) Initially, we select an index, denoted as $b$, using a Bernoulli distribution with a probability of $\frac{1}{2}$. ii) Subsequently, we sample 10 points from the Gaussian distribution $\mathcal{N}(\mu_b, \Sigma)$ to construct the set $S^*$. iii) Further, we sample 90 points for $V \backslash S^*$ from the Gaussian distribution $\mathcal{N}(\mu_{1-b}, \Sigma)$. We follow the procedure of Ou et al. [2022] to obtain 1,000 samples, subsequently divided into training, validation, and test sets. We effectively demonstrate the efficacy of our approach in mastering complex set functions. Detailed results can be found in Table 3.

## 5.2 Product Recommendation

The task involves recommending the most suitable subset of 30 products to a customer within a specific category. For this experiment, we utilize the dataset from the Amazon baby registry, sourced from Gillenwater et al. [2014a]. This dataset includes numerous product subsets chosen by various customers, with Amazon categorizing each item on a baby registry into specific categories such as "Bath", "Health" and "Feeding". Detailed information can be found in Appendix D.

Table 2 presents the performance of all models across different categories. Notably, out of the twelve cases evaluated, HORSE outperforms other models in 11 of them. Even in the Diaper category, our method achieves results that are comparable to INSET, which is noteworthy. These significant improvements highlight the effectiveness and superiority of HORSE. Specifically, while EquiVSet and INSET struggle to surpass classical neural subset selection baselines in the Safety, Car Seats, and Furniture categories, HORSE consistently outperforms all baselines in a notable manner.

## 5.3 Compound Selection in AI-aided Drug Discovery

In drug discovery, the screening of compounds with diverse biological activities and favorable ADME (absorption, distribution, metabolism, and excretion) properties is a critical step [Li et al., 2021, Ji et al., 2022, Gimeno et al., 2019]. Virtual screening typically involves a sequential filtering process that employs multiple essential filters. However, neural networks encounter challenges when learning the complete screening process. This difficulty arises from the absence of intermediate supervision signals, which can be costly or impossible to obtain due to pharmaceutical protection policies. Therefore, we implement a single filter, namely, high bioactivity, to obtain the optimal subset of compound selection, following the methodology in [Ou et al., 2022]. Our experiments are conducted on two datasets: PDBBind [Liu et al., 2015a] and BindingDB [Liu et al., 2007]. To be more practical, we further enhance our approach with the inclusion of *two filters*: the high bioactivity filter and the diversity filter. This extended analysis is denoted as PDBBind-2 and BindingDB-2, representing the two-stage filtering process, for a more practical perspective.

Table 4 demonstrates that our method outperforms the baselines and significantly surpasses random guessing, especially on the BindingDB-2 and PDBBind-2 datasets. However, the improvement on PDBBind and BindingDB is less significant. This marginal enhancement is due to the informative structural characteristics of complexes (the elements within a set), which inherently provide substantial information for this task. Consequently, the model can effectively predict the activity values of

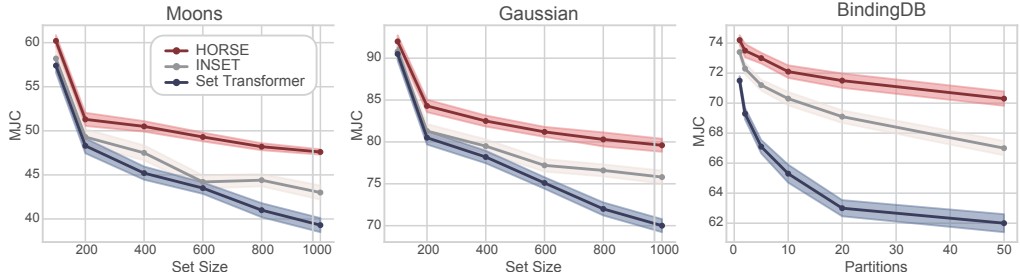

Figure 3: The performance of the methods on the Two-Moons and Gaussian Datasets with respect to the set size is examined in the left two subfigures. These subfigures provide insights into how the performance of the methods varies as the size of the input sets changes. The right subfigure focuses on the influence of the number of partitions on the performance using the BindingDB dataset.

complexes even without explicitly considering the interactions between the optimal subset and its complement in single filter scenarios. Nonetheless, our method still achieves superior results compared to other methods, confirming its effectiveness.

## 5.4 Large-Scale Setting

Set encoding mechanisms, as introduced by Zaheer et al. [2017] and further explored by Lee et al. [2019a], have fundamentally shifted the way neural networks perceive and process sets by emphasizing the importance of permutation invariance and the ability to handle variable-sized inputs. These models are designed to learn from the entire set in a single gradient step, ensuring that the learned representations encapsulate the holistic properties of the set. However, this approach encounters practical limitations when dealing with large-scale sets, where processing the entire set in a single step becomes computationally infeasible due to memory constraints or the sheer volume of data.

To circumvent these challenges, an effective strategy involves training models on partitions of the set, sampled dynamically at each iteration of the optimization process [Lee et al., 2019a, Wang et al., 2024a]. This method allows for manageable subsets to be used for training, significantly reducing the computational load. However, this method will lose information [Bruno et al., 2021, Willette et al., 2023] since it does not process all the elements from $V$. Therefore, we instead partition the set elements into mini-batches, independently encode each batch, and aggregate them to obtain a single set encoding. By applying this methodology across both baseline models and HORSE in scenarios characterized by large-scale input sets, we can highlight the efficiency and scalability of our proposed solution. We conducted experiments on the Two-Moons and Gaussian-Mixture datasets. To ensure consistency, we set the size of the optimal subset $S^*$ to be 10. Subsequently, we varied the size of the input ground set $V$ within the range of $\{200, 400, 600, 800, 1000\}$. Notably, the memory capacity of the GeForce RTX 3090 is insufficient when the size reaches 600. The ground set was divided into 5 disjoint partitions, with each partition containing one-fifth of the elements in $V$. For the purpose of comparison, we selected INSER and Set Transformer as baselines alongside our proposed method, HORSE. INSET demonstrated the best performance among baselines, while Set Transformer is an alternative method that incorporates an attention mechanism. The results obtained from these experiments are presented in the left two subfigures of Figure 3. It is evident that HORSE outperforms both INSER and Set Transformer by a significant margin, demonstrating its superior effectiveness in handling large-scale sets.

Furthermore, to further enhance the practicality of our approach and investigate the potential impact of the partition numbers on the results, we conducted additional experiments on the BindingDB dataset. In this experiment, we set the size of the optimal subset $S^*$ to be 15, while the size of the ground set remained fixed at 1000. We partitioned the ground set into a range of 2 to 50 partitions. The results of these experiments are presented in the right subfigure of Figure 3. Remarkably, it becomes evident that HORSE exhibits remarkable robustness with respect to the number of partitions considered. Regardless of the specific partitioning scheme employed, HORSE consistently delivers exceptional performance, which is more robust than our baselines.

# 6 Conclusion

In this paper, we have addressed the limitations observed in existing methods for neural subset selection tasks. These methods often struggle to effectively model complex information and lack scalability when dealing with large-scale inputs. To overcome these challenges, we propose an innovative and scalable approach called HORSE, which leverages the power of the attention mechanism. Theoretically, we establish that HORSE satisfies the Identity Property and Permutation Invariance, ensuring its soundness and effectiveness. Empirically, we thoroughly evaluate the performance of HORSE against various baselines in both standard and large-scale settings.

*Limitation and Future Work.* Our theoretical and empirical results demonstrate how the attention mechanism can enhance models for neural subset selection tasks in both standard and large-scale settings. However, in large-scale scenarios, our support is currently limited to a theoretical framework for partitioning the set into different groups within a synthetic distributed setting, rather than practical experimentation in a real distributed environment. Moving forward, we plan to implement and test our model in more practical, real-world scenarios to further validate its effectiveness.

# 7 Acknowledgements

We thank all the reviewers for their valuable comments. This work was supported by Research Grants 8601116, 8601594, and 8601625 from the UGC of Hong Kong.

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

# A  Proof

## A.1  Proof of Theorem 4.1

*Proof.* Consider a given set $V = \{x_1, x_2, \ldots, x_n\} \in \mathbb{R}^{n \times d}$, where $n$ represents the number of elements in $V$. Let $\mathbb{S}_n$ represent the set of all permutations of $V$. Now, suppose $\pi_V$ denotes a random permutation applied to $V$. By utilizing this permutation $\pi_V$, we can construct a permutation matrix $M \in \mathbb{R}^{n \times n}$:

$$MV = \begin{bmatrix} -x_{\pi_V(1)}^\top - \\ \vdots \\ -x_{\pi_V(n)}^\top - \end{bmatrix}.$$

Assuming the use of an activation function $\sigma$ that is strictly positive for each element,

$$\sigma(\sqrt{d^{-1}} \cdot q(MVW^K)^\top) = \sigma(\sqrt{d^{-1}} \cdot q(VW^K)^\top M^\top)$$
$$= \sigma(\sqrt{d^{-1}} \cdot qk^\top)M^\top$$
$$= \hat{A}M^\top.$$

The normalized attention score $\text{nl}(\hat{A})$ can be computed using the given permutation $\pi_V$, resulting in

$$\text{nl}(\hat{A}M^\top) = \begin{bmatrix} \hat{A}_{1,\pi(1)}/\sum_{i=1}^k \hat{A}_{i,\pi(1)} & \cdots & \hat{A}_{1,\pi(N)}/\sum_{i=1}^k \hat{A}_{i,\pi(n)} \\ \vdots & \ddots & \vdots \\ \hat{A}_{k,\pi(1)}/\sum_{i=1}^k \hat{A}_{i,\pi(1)} & \cdots & \hat{A}_{k,\pi(n)}/\sum_{i=1}^k \hat{A}_{i,\pi(n)} \end{bmatrix}$$
$$= \text{nl}(\hat{A})M^\top. \tag{13}$$

Now, we consider the matrix multiplication of

$$\text{nl}(\hat{A})M^\top = \begin{bmatrix} \text{nl}(\hat{A})_{1,\pi(1)} & \cdots & \text{nl}(\hat{A})_{1,\pi(n)} \\ \vdots & \ddots & \vdots \\ \text{nl}(\hat{A})_{k,\pi(1)} & \cdots & \text{nl}(\hat{A})_{k,\pi(n)} \end{bmatrix}$$

and

$$MVW^V = \begin{bmatrix} \mathbf{x}_{\pi(1)}^\top W^V \\ \vdots \\ \mathbf{x}_{\pi(n)}^\top W^V \end{bmatrix}.$$

Since

$$\begin{bmatrix} \text{nl}(\hat{A})_{1,\pi(1)} & \cdots & \text{nl}(\hat{A})_{1,\pi(n)} \\ \vdots & \ddots & \vdots \\ \text{nl}(\hat{A})_{k,\pi(1)} & \cdots & \text{nl}(\hat{A})_{k,\pi(n)} \end{bmatrix} \begin{bmatrix} x_{\pi(1)}^\top W^V \\ \vdots \\ x_{\pi(N)}^\top W^V \end{bmatrix}$$

is equal to

$$\begin{bmatrix} \sum_{j=1}^N \text{nl}(\hat{A})_{1,\pi(j)} x_{\pi(j)}^\top W^V \\ \vdots \\ \sum_{j=1}^N \text{nl}(\hat{A})_{m,\pi(j)} x_{\pi(j)}^\top W^V \end{bmatrix},$$

which can also be formulated as:

$$= \begin{bmatrix} \sum_{j=1}^{N} \mathrm{nl}(\hat{A})_{1,j} x_j^\top W^V \\ \vdots \\ \sum_{j=1}^{N} \mathrm{nl}(\hat{A})_{m,j} x_j^\top W^V \end{bmatrix}$$
$$= \mathrm{nl}(\hat{A})v. \tag{14}$$

Therefore, $\mathrm{nl}(\hat{A})v$ is permutation invariant under the permutation group of $\mathbb{S}_n$. Since

$$\mathrm{nl}(\hat{A})\mathbf{1}_n = \sum_{j=1}^{n} \mathrm{nl}(\hat{A})_{i,j} = \sum_{j=1}^{N} \mathrm{nl}(\hat{A})_{i,\pi_V(j)},$$

thus $\mathrm{diag}\left(\mathrm{nl}(\hat{A})\mathbf{1}_n\right)^{-1}$ is also invariant with respect to the permutation of input $V$, which leads to the conclusion that

$$h(MV) = h(V).$$

Similarly, we can construct the permutation matrix $M \in R^{s\times s}$ for a given $S$ and permutatation $\pi_S$, such that:

$$MS = \begin{bmatrix} -x_{\pi_S(1)}^\top - \\ \vdots \\ -x_{\pi_S(s)}^\top - \end{bmatrix}.$$

with the same process as Equation 13 and 14, we can have the following conclusion:

$$F(h(S), h(V)) = F(h(\pi_S \cdot S), h(\pi_V \cdot V)).$$

$\square$

## B  Proof of Theorem 4.2

*Proof.* Consider the input set $V \in \mathbb{R}^{n\times d}$ and let $V = S_1 \cup S_2 \cup \cdots \cup S_m$ represent a partition of $V$ with $|S_i| = n_i$. In other words, $V$ can be expressed as the union of all $S_i$ and each $S_i$ is disjoint from $S_j$ for $i \neq j$. Without loss of generality, we can make the assumption that.

$$k = VW^k = \begin{bmatrix} S_1 W^k \\ \vdots \\ S_m W^k \end{bmatrix}, \quad v = VW^v = \begin{bmatrix} S_1 W^v \\ \vdots \\ S_m W^v \end{bmatrix}$$

where $S_i W^k \in \mathbb{R}^{n_i \times d_h}$ and $S_i W^v \in \mathbb{R}^{n_i \times d_h}$ for $i = 1, 2 \ldots, m$. Then we can express the matrix $\mathrm{nl}(\hat{A})$ as follows:

$$\mathrm{nl}(\hat{A}) = \left[ \mathrm{nl}(\hat{A}^{(1)}) \cdots \mathrm{nl}(\hat{A}^{(m)}) \right],$$

where $\hat{A}^{(i)} = \sigma(\sqrt{d^{-1}} \cdot q(S_i W^k)^\top) \in \mathbb{R}^{m\times n_i}$ for $i = 1, 2 \ldots, m$ since $\mathrm{nl}(\hat{A})_{i,j}$ is independent to $\mathrm{nl}(\hat{A})_{i,t}$ for all $t \neq j$.

Since

$$\bar{h} = \left[ \mathrm{nl}(\hat{A}^{(1)}) \cdots \mathrm{nl}(\hat{A}^{(m)}) \right] \begin{bmatrix} S_1 W^v \\ \vdots \\ S_m W^v \end{bmatrix},$$

the following equality holds

$$\bar{h} = \sum_{i=1}^{m} \text{nl}(\hat{A}^{(i)}) S_i W^v \qquad (15)$$

Since

$$M_i = \sum_{t=1}^{m} \sum_{j=1}^{n_i} \text{nl}(\hat{A}^{(t)})_{i,j},$$

we can decompose $\text{nl}(\hat{A})\mathbf{1}_n$ as

$$\begin{aligned}
\text{nl}(\hat{A})\mathbf{1}_n &= \sum_{i=1}^{m} \left( \sum_{j=1}^{n_i} \text{nl}(\hat{A}^{(i)})_{1,j}, \ldots, \sum_{j=1}^{n_i} \text{nl}(\hat{A}^{(i)})_{m,j} \right)^{\top} \\
&= \sum_{i=1}^{m} \text{nl}(\hat{A}^{(i)})\mathbf{1}_{n_i}
\end{aligned} \qquad (16)$$

where $\mathbf{1}_{n_i} = (1, \ldots, 1) \in \mathbb{R}^{n_i}$. Combining Equation 15 and 16

$$h(V) = \text{diag}\left( \sum_{i=1}^{m} \text{nl}(\hat{A}^{(i)})\mathbf{1}_{n_i} \right)^{-1} \left( \sum_{i=1}^{m} \text{nl}(\hat{A}^{(i)}) S_i W^v \right).$$

We define $h_1(S_i) = \text{nl}(\hat{A}^{(i)})\mathbf{1}_{n_i}$ and $h_2(S_i) = \text{nl}(\hat{A}^{(i)}) S_i W^v$. Moreover, $h(S_i) = \text{diag}(h_1(S_i))^{-1} h_2(S_i)$ Now, we define a function,

$$\begin{aligned}
g(\{h(S_1), \ldots, h(S_m)\}) &:= \\
g_1(\{h_1(S_1), \ldots, h_1(S_m)\}) &\cdot g_2(\{h_2(S_1), \ldots, h_2(S_m)\}),
\end{aligned}$$

where

$$g_1(\{h_1(S_1), \ldots h_1(S_m)\} := \text{diag}\left( \sum_{i=1}^{m} h_1(S_i) \right)^{-1}$$

$$g_2(\{h_2(S_1), \ldots, h_2(S_m)\}) := \sum_{i=1}^{m} h_2(S_i).$$

Then $h(V) = g(\{h(S_1), \ldots, h(S_m)\})$. Since the partition is arbitrary, $h$ satisfies Property 3.1.

$\square$

## C   Pseudo Code of `HORSE`

In the main text, we present `HORSE` using matrix calculations, which may be challenging to comprehend. To improve understanding of how our method establishes an attention map between subsets $S_i$, we detail the procedural steps in Algorithm 1, with a special emphasis on the generation of $h(S_i)$. This approach is designed to elucidate the operational details and highlight the methodology involved in constructing $h(S_i)$.

---

**Algorithm 1** HORSE. $V = \{S_1, S_2, \ldots, S_m\}$ is the input set partitioned into $m$ subsets. $\xi \in \mathbb{R}^{m \times d_s}$ is the initialized embedding and $g$ is the choice of aggregation function.

---

1: **Input:** $V = \{S_1, S_2, \ldots, S_m\}, S = S_1, \zeta \in \mathbb{R}^{m \times d_s}, g$
2: **Output:** $Z \in \mathbb{R}^{d_o}$
3: **Initialize** $\zeta$
4: $q = LN(\zeta W^q)$
5: **for** $i = 1, 2, \ldots, m$ **do**
6:     $k_i = S_i W^k$
7:     $v_i = S_i W^v$
8: **end for**
9: $k = [k_1, k_2, \ldots, k_m]^T$
10: $v = [v_1, v_2, \ldots, v_m]^T$
11: $\hat{M} = \sqrt{d_h^{-1} \cdot qk^T}$
12: $\hat{A} = \sigma(M_1)$
13: $A = \text{nl}(\hat{A}) = [A_1, A_2, \ldots, A_m]^T$
14: $M = diag(\text{nl}(A)\mathbf{1}_n)^{-1} = [M_1, \ldots, M_m]^T$
15: **for** $i = 1, 2, \ldots, m$ **do**
16:     $h(S_i) = M_i A_i S_i W^v$
17: **end for**
18: $\hat{S}_i = g(h(S_1), \ldots h(S_m))$
19: $Z = F(h(S), \hat{S}_i)$
20: **return** $Z$

---

## D   Experimental Details and Additional Experiments

### D.1   Detailed Description of Tasks.

**Product Recommendation.**  The task involves recommending the most suitable subset of 30 products to a customer within a specific category. For this experiment, we utilize the dataset from the Amazon baby registry, sourced from Gillenwater et al. [2014a]. This dataset includes numerous product subsets chosen by various customers, with Amazon categorizing each item on a baby registry into specific categories such as "Bath", "Health" and "Feeding". Additionally, each product is represented by a 768-dimensional vector generated by a pre-trained BERT model, based on its textual description.

The Amazon baby registry data [Gillenwater et al., 2014b] comprises various datasets collected from Amazon, encompassing different categories such as toys, furniture, and more. Within each category, there exist $|V|$ sets of products that have been selected by different customers. To create a sample $(S^*, V)$, we follow a specific procedure. Initially, we remove any subset with an optimal subset size $|S^*|$ greater than or equal to 30. The remaining subsets are then divided into training, validation, and test folds using a 1:1:1 ratio. Additionally, we randomly select an additional $30 - |S^*|$ products from the same category to construct $(S^*, V)$. This process allows us to create a data point $(S^*, V)$. For comprehensive information, please refer to Table 5 in Ou et al. [2022], which presents the statistics of the categories.

**Set Anomaly Detection.**  We tackle set anomaly detection tasks on four real-world datasets: double MNIST [Sun, 2019], CelebA [Liu et al., 2015b], F-MNIST [Xiao et al., 2017], and CIFAR-10 [Krizhevsky, 2009]. Each dataset is partitioned into training, validation, and test sets, each comprising

Table 5: The statistical properties of the Amazon product dataset.

| Categories | $|\mathcal{D}|$ | $|V|$ | $\sum |S^*|$ | $\mathbb{E}[|S^*|]$ | $\min_{S^*} |S^*|$ | $\max_{S^*} |S^*|$ |
|---|---|---|---|---|---|---|
| Gear | 4,277 | 30 | 16,288 | 3.80 | 3 | 10 |
| Bath | 3,195 | 30 | 12,147 | 3.80 | 3 | 11 |
| Toys | 2,421 | 30 | 9,924 | 4.09 | 3 | 14 |
| Media | 1,485 | 30 | 6,723 | 4.52 | 3 | 19 |
| Safety | 267 | 30 | 846 | 3.16 | 3 | 5 |
| Diaper | 6,108 | 30 | 25,333 | 4.14 | 3 | 15 |
| Health | 2,995 | 30 | 11,053 | 3.69 | 3 | 9 |
| Carseats | 483 | 30 | 1,576 | 3.26 | 3 | 6 |
| Bedding | 4,524 | 30 | 17,509 | 3.87 | 3 | 12 |
| Feeding | 8,202 | 30 | 37,901 | 4.62 | 3 | 23 |
| Apparel | 4,675 | 30 | 21,176 | 4.52 | 3 | 21 |
| Furniture | 280 | 30 | 892 | 3.18 | 3 | 6 |

10,000, 1,000, and 1,000 samples, respectively. In each dataset, we randomly select $n$ images from the dataset to create the OS Oracle $S^*$, where $n$ can be either 2, 3, or 4. This setup aligns with the approach outlined in [Zaheer et al., 2017, Ou et al., 2022].

Let's take CelebA as an illustrative example. In this scenario, the goal is to identify anomalous faces solely through visual observation, without using any attribute values. The CelebA dataset consists of 202,599 face images, each annotated with 40 boolean attributes. When constructing sets, for each ground set $V$, we randomly select $n$ images from the dataset to create the OS Oracle $S^*$, ensuring that none of the selected images contain any of the two attributes. Additionally, we ensure that no individual person's face appears in both the training and test sets.

Regarding the results presented in Table 6, it is evident that our model exhibits a significant performance advantage over all the baseline methods. This substantial improvement underscores the superior capabilities of our model in addressing the given task.

**Compound Selection in AI-aided Drug Discovery.** In drug discovery, the screening of compounds with diverse biological activities and favorable ADME (absorption, distribution, metabolism, and excretion) properties is a critical step [Li et al., 2021, Ji et al., 2022, Gimeno et al., 2019]. Virtual screening typically involves a sequential filtering process that employs multiple essential filters. These filters initially select diverse subsets from highly active compounds and subsequently eliminate compounds with unfavorable ADME characteristics. After passing through several filtering stages, an optimal subset of compounds is identified. However, neural networks encounter challenges when learning the complete screening process. This difficulty arises from the absence of intermediate supervision signals, which can be costly or impossible to obtain due to pharmaceutical protection policies. Consequently, models are expected to learn this intricate selection process in an end-to-end manner. In other words, models must predict $S^*$ based solely on the optimal subset supervision signals, without knowledge of the intermediate steps. Therefore, we simulate the optimal subset oracle of compound selection by applying one or two filters by uising PDBBind and BindingDB, as [Ou et al., 2022].

PDBBind offers an extensive compilation of experimentally measured binding affinity data for biomolecular complexes. We utilized the "refined" portion of the complete PDBBind dataset, which consists of 179 complexes, to construct our subsets. To create a data point $(V, S^*)$, we randomly sampled 30 complexes from the dataset to form the ground set $V$. The subset $S^*$ was then generated by selecting the five most active complexes within $V$. We constructed separate training, validation, and test splits, comprising 1000, 100, and 100 data points, respectively.

BindingDB is an openly accessible database that provides measured binding affinities for a collection of $52,273$ drug-target pairs involving small, drug-like molecules. Similar to PDBBind, we randomly selected 300 drug-target pairs from the BindingDB database to form the ground set $V$. From this dataset, we carefully identified the 15 most active drug-target pairs and designated them as $S^*$. To ensure comprehensive evaluation and robust model training, we subsequently created distinct training, validation, and test sets. These comprised 1000, 100, and 100 data points respectively.

Table 6: Empirical results of set anomaly detection Tasks. Bolded numbers denote the best performance. HORSE outperforms all the baselines on the four datasets.

| | Random | PGM | DeepSet | Set-T | EquiVSet | INSET | HORSE |
|---|---|---|---|---|---|---|---|
| Double MNIST | 8.2 | 30.0±1.0 | 11.1±0.3 | 51.2±0.5 | 57.5±1.8 | 69.7±1.0 | **72.3 ± 1.2** |
| CelebA | 2.2 | 48.1±0.6 | 44.0±0.6 | 52.7±0.8 | 54.9±0.5 | 57.5±1.2 | **59.3 ± 1.0** |
| F-MNIST | 1.9 | 54.0±2.0 | 49.0±2.0 | 58.1±1.0 | 65.0±1.0 | 70.1±2.1 | **73.5 ± 1.6** |
| CIFAR-10 | 1.9 | 45.0±2.0 | 32.0±0.8 | 65.0±2.3 | 60.0±1.2 | 71.2±2.1 | **74.3 ± 1.2** |
| Average | 3.55 | 44.28 | 34.03 | 56.75 | 59.35 | 67.13 | **69.85** |

## D.2 Descriptions of Baselines

*Random.* This represents the expected performance of a random guess, serving as a baseline to help us gauge the actual difficulty of the tasks.

*PGM [Tschiatschek et al., 2018].* PGM, which stands for Probabilistic Greedy Model, tackles the optimization Problem 1 by employing a differentiable extension of the greedy maximization algorithm. For a deeper understanding of this approach, please refer to the origianl paper or Appendix A in [Ou et al., 2022].

*DeepSet [Zaheer et al., 2017].* Here, we employ DeepSet as a baseline model. We use DeepSet to predict the probabilities of including specific instances in $S^*$, essentially learning an invariant permutation mapping from the power set $2^V$ to the interval $[0, 1]$ of size $|V|$. This model was also used as a backbone for set function learning in EquiVSet. Moreover, it is suitable for subset selection tasks, as detailed in its original paper.

*Set Transformer [Lee et al., 2019a].* Set Transformer extends DeepSet's capabilities by integrating the self-attention mechanism. This addition allows the model to consider pairwise interactions between elements, enabling it to capture dependencies and relationships among different elements more effectively. It can be also utilized for subset selection tasks, similar to DeepSet.

*EquiVSet [Ou et al., 2022].* EquiVSet employs an energy-based model (EBM) to establish the set mass function, denoted as $P(S|V)$ from a probabilistic standpoint. Their primary objective lies in learning a distribution $P(S|V)$ that monotonically increases with respect to the utility function $F(S, V)$. It's worth noting that their framework focuses on approximating the symmetric function $F(S)$ rather than the symmetric function $F(S, V)$, with DeepSet serving as the foundational component of their model to approximate the set function.

*INSET [Xie et al., 2024].* As discussed in the Introduction, EquiVSet faces limitations in incorporating information from the ground set $V$. In response to this challenge, Xie et al. [2024] present an innovative solution. They propose the generation of embeddings for $V$ and subsequently concatenate these embeddings with the representations of $S$, which has been presented in Figure 1.

Among these baselines, DeepSets and Set Transformer are two crucial model structures widely used in set-based tasks, including subset selection tasks. On the other hand, PGM, EquiVSet, and INSET are methods specifically designed for neural subset selection tasks. Notably, both INSET and HORSE are implemented based on the EquiVSet framework, yet they significantly outperform EquiVSet.

## D.3 The Objective of Neural Subset Selection in Optimal Subset Oracle

Our method, HORSE, is applicable for learning $F(S, V)$ across a range of tasks. In this paper, we primarily utilize the framework established in [Ou et al., 2022] and modify it with our approach to model $F(S, V)$. The optimization objective aims to solve Equation 1 by employing an implicit learning strategy based on probabilistic reasoning. This approach can be formulated concisely as follows:

$$\arg\max_{\theta} \; \mathbb{E}_{\mathbb{P}(V,S)}[\log p_\theta(S^*|V)]$$
$$\text{s.t.} \; p_\theta(S|V) \propto F_\theta(S; V), \forall S \in 2^V,$$

Table 7: In the table, we report the performance of different sample numbers denoted by "k" and compare them against the best-performing baselines.

|  | Media | Safety |
|---|---|---|
| Best Baseline | $62.0 \pm 2.3$ | $25.0 \pm 0.6$ |
| k=2 | $63.1 \pm 1.2$ | $24.8 \pm 1.3$ |
| k=4 | $64.4 \pm 1.0$ | $25.9 \pm 1.2$ |
| k=6 | $65.8 \pm 1.2$ | $26.8 \pm 0.9$ |
| k=8 | $\mathbf{66.8 \pm 1.3}$ | $27.4 \pm 1.0$ |
| k=10 | $66.2 \pm 0.9$ | $\mathbf{27.7 \pm 0.8}$ |

Constructing a suitable set mass function $p_\theta(S|V)$ that exhibits monotonicity with respect to the utility function $F_\theta(S; V)$ is a crucial aspect of tackling this problem. To accomplish this, we can utilize the Energy-Based Model (EBM):

$$p_\theta(S|V) = \frac{\exp(F_\theta(S; V))}{Z}, \; Z := \sum_{S' \subseteq V} \exp(F_\theta(S'; V)),$$

In practice, we approximate the Energy-Based Model (EBM) through a variational approximation. Due to the scope of this paper, we omit the detailed explanation for the sake of simplicity. We kindly invite readers to refer to [Ou et al., 2022] for further information on this topic.

### D.4 Implementation Details

In this subsection, we present the implementation details of HORSE. The setup closely follows that of [Ou et al., 2022]. The proposed models are trained using the Adam optimizer [Kingma and Ba, 2014] with a fixed learning rate of $1e-4$ and a weight decay rate of $1e-5$. To accommodate different model sizes across various datasets, we select the batch size from the set $\{4, 8, 16, 32, 64, 128\}$. Importantly, we choose the largest batch size that allows efficient training on a single GeForce RTX 3090 GPU.

To enhance training efficiency and mitigate overfitting, we utilize an early stopping strategy for both the baselines and our proposed models. In this strategy, if there is no performance improvement over $10$ consecutive epochs, we terminate the training process prematurely. For each dataset, the maximum number of epochs allowed for training is set to $80$. At the end of each epoch, we assess the model's performance on the validation set and save the model with the best performance. Finally, we evaluate the saved models on the test set to determine their performance.

In order to consider the effect of randomness and ensure the reliability of the findings, we conduct all experiments five times using different random seeds. The average performance metrics, along with their corresponding standard deviations, are reported as the final performance measures. This approach allows for a comprehensive evaluation of the models' performance while accounting for the influence of random variations.

### D.5 Ablation Study

To further investigate the robustness of INSET, we have conducted ablation studies specifically focusing on the Monte-Carlo sample numbers $k$ for each input pair $(S^*, V)$. In our framework, the model $\theta$ is trained to accurately predict the optimal subset $S^*$ from a given ground set $V$. Following the Energy-Based Method (EBM) proposed in [Ou et al., 2022], we incorporate a necessary hyper-parameter. During the training process, we sample $k$ subsets from $V$ in order to optimize the model parameters $\theta$, thereby maximizing the conditional probability distribution $p_\theta(S^*|V)$ among all pairs of $(S, V)$ for a given $V$.

To evaluate the robustness of HORSE across various values of $k$, we perform experiments on Media and Safety Categories of Product Recommendation. The results of these experiments are presented in Table 7, providing a comprehensive overview of the performance achieved with different $k$. Our findings clearly demonstrate that HORSE consistently outperforms all other baselines across the entire range of $k$ values considered.

