# OpenReview forum: "HORSE: Hierarchical Representation for Large-Scale Neural Subset Selection"
_NeurIPS.cc/2024/Conference — NeurIPS 2024 poster_

### Official Review · Reviewer_Unf9 · 2024-07-13

**Soundness:** 3
**Presentation:** 3
**Contribution:** 3
**Rating:** 7
**Confidence:** 4

**Summary:**

The authors propose a novel approach for architecture for the task of subset selection that is based on developing layers (HORSE) which satisfy a notion of an identity property (leveraging learning representations that connect a set V with its subset S) in a permutationally-invariant way, leveraging key-query-based attention mechanisms. Extensive experiments and theoretical justification validate the proposed construction.

**Strengths:**

I appreciate the clear presentation, and the introduction of the Identity Property. The work is very well presented and the experiments show clear improvements of HORSE upon INSET and Set Transformers.

**Weaknesses:**

Unless I have missed it, could you let me know what is the effect of the random partitioning of the set V? Do you have ablations over that? How sensitive is your method to the partitioning?

nit: In Figure 2's caption you have a typo: if the ground set <- of the ground set.

nit: In Table 4 you may want to use '' instead of " for a closing quotation mark.

**Questions:**

How do you define an "effective" aggregation function in Property 3.1? In principle, by this definition, $g$ can be a trivial function such as $g \equiv 0$, which will ignore all the subsets $S_1,\dots,Sm$.

**Limitations:**

Yes.

---

> ### Author Rebuttal · Authors · 2024-08-07
>
> Thank you very much for appreciating our work and helping us correct the typos! We would like to address your insightful questions with the following responses.
>
> ***What is the role of the random partitioning of the set V?***
>
> We consider scenarios where the ground set V is so large that a single GPU cannot process all elements simultaneously. Consequently, we must partition $V$ into various partitions, $S_1, S_2, \dots, S_m$, to obtain individual representations and subsequently combine them. We suggest that methodologies should be capable of processing and aggregating each subset from a set partition, yielding an identical representation as if the entire set was encoded at once (**Identity Property**). For clarity, this can be expressed as f(V) = $g(f(S_1), f(S_2), \dots, f(S_m))$, where $m$ can vary while remaining less than or equal to $n$. According to Theorem 4.2, our methods are designed to support the random partitioning of the set $V$.
>
> ***What is the impact of the random partitioning in the experiments?***
>
> In our theorems, we do not impose any limitations on the size or number of partitions, allowing for arbitrary partitioning. In practical terms, we use two partitions (m=2) when comparing HORSE with the baselines. Furthermore, we present Figure 3 to demonstrate the impact of varying the number of partitions and the size of each partition on the results. While all neural subset methods experience a decline in performance as the number of partitions increases, HORSE demonstrates significantly **more robust and superior performance**. This implies that HORSE is more robust in large-scale settings.
>
> To provide further evidence and for your convenience, we have included new results on BindingDB-2 in the table below. Please note that for simplicity, we maintain an equal size for each partition. Otherwise, there would be numerous possible partitioning methods, making experimentation challenging.
>
> |   | m=2 |  m=3          | m=5          | m=6          | m=10                     | m=15             |
> | ----------- | -------------- | -------------- | -------------- | -------------- | ------------------------ | ------------------------ |
> | INSET      | 0.164±0.010 | 0.152±0.017 | 0.133±0.025 | 0.135±0.027 | 0.118±0.041 | 0.109±0.062 |
> | Set-T   | 0.149±0.016 | 0.134±0.033 | 0.121±0.032 | 0.122±0.039 | 0.109±0.038           | 0.101±0.045 |
> | HORSE      | **0.213 ±0.005** | **0.197±0.015** | **0.163±0.023** | **0.160±0.029** | **0.124±0.043**           | **0.115±0.055** |
>
>
> ***How to define the "effective" aggregation function in Property 3.1?***
>
>  The aggregation function can be calculated based on our proof of Theorem 4.2. Specifically, for our method HORSE, the expression of g is defined in the following formula:
>
> $$
> g(\{h(S_1), \dots, h(S_m) \} ) \coloneqq  \text{diag}\left(\sum_{i=1}^m h_1(S_i)\right)^{-1}  \cdot \sum_{i=1}^m h_2(S_i),
> $$
>
> where $h_1(S_i) = nl(\hat{A}^{(i)})\mathbb{1}_{n_i}$ and $h_2(S_i) = nl(\hat{A}^{(i)})S_iW^v.$ The algorithm $h(S_i)$ can be found in our Algorithm 1. In practice, due to the neural network's ability to learn useful $h_1(S_i)$ and $h_2(S_i)$, set-related works typically employ pooling methods as the aggregation method, e.g., [1,2,3,4]. In our work, we use the most common method – mean pooling. We also conducted ablation studies on the pooling methods to demonstrate the effectiveness of the aggregation methods. These four methods have similar overall performance, with sum pooling and mean pooling being slightly better.
>
> |   | mean pooling |  sum pooling          | min pooling          | max pooling
> | ----------- | -------------- | -------------- | -------------- | -------------- |
> | PDBBind      | **0.941 ± 0.007** | 0.939±0.010 | 0.934±0.013 | 0.935±0.012 |
> | BindingDB   | 0.742 ± 0.011 | **0.745±0.012** | 0.739±0.015 | 0.736±0.019 |
> | PDBBind-2      | 0.432 ± 0.006 | **0.435±0.008** | 0.423±0.009 | 0.425±0.012 |
> | BindingDB-2      | **0.213 ± 0.009** | 0.210±0.007 | 0.211±0.011 | 0.206±0.014 |
>
> We sincerely thank you once again for your time and valuable contribution. Should you have any additional suggestions or questions, please do not hesitate to let us know.
>
> ---------------
> [1] Maron et al., "On learning sets of symmetric elements," ICML, 2020.
>
> [2] Zaheer et al., "Deep Sets", NIPS 2017.
>
> [3] Willette et al., "Universal Mini-Batch Consistency for Set Encoding Functions", arXiv 2022.
>
> [4] Xie et al., "Enhancing Neural Subset Selection: Integrating Background Information into Set Representations", ICLR 2024.

---

> > ### Comment · Reviewer_Unf9 · 2024-08-11
> > **Thanks for the rebuttal.**
> >
> > The authors have addressed my comments, I have read the discussions, and maintain my score.

---

> > > ### Author Response · Authors · 2024-08-11
> > >
> > > Thank you for your response and for dedicating your time to thoroughly review our paper and rebuttal.

---

### Official Review · Reviewer_6Cmw · 2024-07-17

**Soundness:** 3
**Presentation:** 4
**Contribution:** 3
**Rating:** 6
**Confidence:** 3

**Summary:**

The paper studies an interesting and important problem that is general in machine learning, focusing on motivations (the coexistence of the set-to-set interaction and the large-scale setting issues) that are crucial in this field. While people may like to see the real-world practical analysis of this method, the theoretical analysis and experiments show that the proposed method is effective in the field of recommendation and anomaly detection. Also, the paper is well written and easy to understand.

**Strengths:**

1. The paper studies an interesting and important problem that is general in machine learning. The problem, if effectively solved, can be used in a variety of ML tasks.
2. Motivations of this paper, the coexistence of the set-to-set interaction and the large-scale setting issues, are crucial in this field.
3. The paper's writing is good. It is informative as well as easy to understand. Explanations of motivations and methodology insights are straightforward but useful.

**Weaknesses:**

1. In Section 3.1, I cannot find citations for the problem setting. At first glance, it seems the problem setting does not reflect real cases. Why is $W$ split into $$ m"disjoint" subsets a given assumption? If the subsets are provided as candidates, what if they limit the results to suboptimal solutions or introduce bias?
2. More discussion is needed regarding the impacts of $n_i$ and $m$. Although they are given as fixed inputs, it is possible to merge some subsets, which $m$ decreases and $n_i$ increases, before running the algorithm.
3. The proposed method has been tested under anomaly detection and recommendation tasks. However, it has not been compared with popular approaches in these fields that do not use subset-valued functions. The method has only been compared with others within the set-function learning domain. While this is necessary, it might be a minor weakness given that the author has stated their model is theoretical rather than practical for real-world scenarios.

**Questions:**

1. Can the proposed framework be used in the image pixel grouping task, such as image segmentation? Does predicting a subset of pixels from the entire image pixels fit in the subset-valued function?
2. Why splitting V into "disjoint" subsets is necessary? Why not just randomly sampling subsets from V such that different subsets may overlap?

**Limitations:**

The authors addressed the limitations in the conclusion section.

---

> ### Author Rebuttal · Authors · 2024-08-07
>
> Thank you very much for your appreciation of our work and for posing such interesting and promising questions! We will first offer some clarifications and then proceed to response to your insightful questions.
>
> ***Some confusion regarding the experimental settings***
>
> As stated in the Introduction, the framework of our problem is derived from references [1,2]. For tasks involving neural subset selection, all existing methodologies necessitate the construction of models to learn a set function, $F(S,V)$. Our enhancement to [1, 2] involves considering scenarios where the ground set V is so large that a single GPU cannot process all elements simultaneously. Consequently, we must partition V into various sections, $S_1, S_2, \dots, S_m$, to obtain individual representations and subsequently combine them.
>
> We suggest that methodologies should be capable of processing and aggregating each subset from a set partition, yielding an identical representation as if the entire set was encoded at once (Identity Property). For simplicity, this can be expressed as $f(V) = g(f(S_1), f(S_2), …, f(S_m)$, where m can vary while remaining less than or equal to $n$.
>
> In our theorems, we do not impose any limitations on the size or number of partitions, meaning the partition can be arbitrary.  In practical terms, we use two partitions (m=2) when comparing HORSE with the baselines. Moreover, we present Figure 3 to demonstrate how variations in $m$ and $n_i$ will impact the results. While all neural subset methods will experience a drop in performance, HORSE exhibits significantly more **robust and superior performance**. This implies that HORSE will be more robust in large-scale settings.
>
> To provide further evidence and for your convenience, we have included new results on BindingDB-2 in the table below. Please note that for simplicity, we ensure that each partition has an equal size by maintaining the same value of $n_i$ across all partitions. Otherwise, there would be numerous possible partitioning methods, making experimentation challenging.
>
> |   | m=2 |  m=3          | m=5          | m=6          | m=10                     | m=15             |
> | ----------- | -------------- | -------------- | -------------- | -------------- | ------------------------ | ------------------------ |
> | INSET      | 0.164±0.010 | 0.152±0.017 | 0.133±0.025 | 0.135±0.027 | 0.118±0.041 | 0.109±0.062 |
> | Set-T   | 0.149±0.016 | 0.134±0.033 | 0.121±0.032 | 0.122±0.039 | 0.109±0.038           | 0.101±0.045 |
> | HORSE      | **0.213 ±0.005** | **0.197±0.015** | **0.163±0.023** | **0.160±0.029** | **0.124±0.043**           | **0.115±0.055** |
>
> ***Can the proposed framework be used in the image pixel grouping task, e.g., image segmentation and predicting a subset of pixels from the entire image pixels?***
>
> In Neural Subset Selection tasks, the central premise is the necessity to identify an optimal subset for each ground set. In other words, we can only predict one optimal subset S* from one set V. In contrast, determining an optimal subset for image segmentation is challenging, as it is difficult to pinpoint which part is optimal. Nonetheless, predicting a subset of pixels (that satisfy certain properties) appears to align with the objectives of neural subset selection tasks. In this case, HORSE can be potentially applied, as pixels can serve as elements and the entire image constitutes the ground set. The primary challenge when employing neural subset selection methods might be encoding each pixel into a useful embedding, which we consider as a very promising future extension of our methods.
>
> ***Why is the partitions required to be disjoint?***
>
> Our ultimate objective is to enable methods to process and aggregate each subset from a set partition, producing a representation identical to encoding the entire set at once. Intuitively, if the subsets overlap, aggregating their representations would introduce redundant information, making it challenging for models to identify the true ground set V. This difficulty arises as models may not discern that $ \\\{ e_1, e_2, e_3, e_4 \\\} $ and $\\\{ e_1, e_2, e_2, e_3, e_3, e_4 \\\}$ represent the same set. Thus, overlapping subsets can pose additional challenges for model learning. For instance, given $V = \\\{e_1, e_2, e_3, e_4\\\}$, it is easier to train a model to satisfy $f(V) = g(f(\\\{e_1, e_2\\\}), f(\\\{e_3, e_4\\\}))$ than to train a model to satisfy $f(V) = g(f(\\\{e_1, e_2, e_3\\\}), f(\\\{e_2, e_3, e_4\\\}))$. The second method required the models to possess the capability to disregard the repeated elements.
>
> Thank you for your time and thoughtful consideration again. If you have any concerns or questions, please don't hesitate to reach out to us.
>
> -------------------------
> [1] Ou et al., “Learning neural set functions under the optimal subset oracle” NeurIPS 2022.
>
> [2] Tschiatschek et al., “Differentiable submodular maximization,” IJCAI, 2018.

---

### Official Review · Reviewer_aVVy · 2024-07-22

**Soundness:** 3
**Presentation:** 1
**Contribution:** 2
**Rating:** 6
**Confidence:** 4

**Summary:**

Paper is interested in subset selection. Given ground-set $V$, how to choose a subset $S \subseteq V$ that maximizes utility function $F(S, V)$. The core contribution of the method is in (randomly) partitioning $V$ into multiple subsets $S_1, S_2, \dots, S_m$, computing some matrices (based on this random partitioning), and using these matrices to compute information from $S$ and each of $S_i$, plugging into neural network, that should model $F$.

**Strengths:**

* The direction of the paper is interesting: mapping variable-sized lists, through a neural network.
* The random partitioning works well when the size of sets is large ($m$ can be chosen to be larger, if the ground set is larger).
* Good experimental results.

**Weaknesses:**

In my opinion, there are two main weaknesses in the paper:

1. The "subset formulation" seems to be introduced by force (I don't see a good fit here).
2. The notation is **unnecessarily** complex.

## Subset Formulation

Usually, for subset-selection (IMO), there are usually some conditions along:

1. Cardinality constraint. E.g., $|S| \le k$. In such applications, enlarging $S$ is usually good -- almost every item gives some positive utility, however, some give utility more than others  -- e.g., submodular.

2. The presence of some items makes other items less-needed. E.g., if a customer order contains diapers from company A then they get less if buying diapers from company B.

However, the experiments (in my understanding) seem to be classification-like or detection-like -- they are more fit for multi-class classification. This becomes ecident as the qualitative measure of Table 1 experiments is Jaccard similarity (it strictly grows when including correct elements and degrades when including incorrect elements).

NOTE: this does *not* degrade any of the modeling tricks they have invented in the paper, I am only complaining about advertising the wrong thing (per my understanding, and please correct my understanding if I am wrong).

## Unnecessary complexity

The reason for notation is conciseness and clarity. I think one should spend a lot of time on carefully-choosing notation to carry the message clearly. In cases where notation is complicating things, it is defeating its own purpose. I will point some specifics here.

* The calculation of $h(V)$ seems to be an algorithm -- specifically, detailed in section 3.3  -- roughly, compose
matrix $\zeta$ based on random partitioning of rows of $V$, from which, matrices (q, v, k) are calculated which finally produce the output of $h(V)$. The notation in Equation (11) does not use $h(V)$ but instead uses $h(S)$ and $h(S_i)$. In that case, would the $\zeta$ be recalculated for every invocation of $h$ or is it computed from the ground-set $V$? If the latter, it might make sense to denote $h_V(S)$.

* matrix $A$ is double (L1) normalized: Eq.8 normalizes columns to sum to `1` and Eq.9 (and its application in Eq.10) normalizes rows to sum to `1` -- but why have completely different notations to the normalization?

* Note: the wording "utilizing unnormalized" scores made me think that it might be a good idea to do "sum" rather than "mean" (i.e., unnormalized pooling). However, the matrix $A$ is indeed double normalized.

The letter $n$ denotes number of nodes, yet $nl$ stands for normalization. I recommend you use \textrm{nl} for the function call.

* Set are permutation invariant by construction. This makes Theorem 4.1 unnecessary. However, I understand your intent: you are referring to the matrix version and not the set. Please correct the writing.

* Please be clear on the permutation's $\pi$. It seems that it is a binary matrix where every row and every column has exactly single 1.

**Questions:**

Q1: What is the objective function that you used for training? While you are focusing on the model, it is especially important to note it, as the method is advertised as "subset selection" from the start.

Q2: How is row $i$ of $\zeta$ calculated from points clustered in $S_i$. Is it the average? This crucial information is missing.

Q3: Do h(V) and h(S) share the same $\zeta$?

**Limitations:**

Authors have added this as part of their conclusion

---

> ### Author Rebuttal · Authors · 2024-08-07
>
> We sincerely appreciate the time and effort you have invested! Your suggestions regarding notations and typos have greatly assisted us in revising our manuscript. For the remaining concerns, we offer the corresponding clarifications below.
>
> ***The difference between neural subset selection and (core) subset selection.***
>
> As described in our Introduction, the configuration of our problem primarily originates from references [1, 2], which postulate that the underlying process of determining the optimal $S^*$ can be modeled by a  utility function $F_\theta (S;V)$ parameterized by $\theta$,  and the following criteria:
>
> $$
>     S^* = argmax_{S \in 2^{V}} F_\theta (S; V).
> $$
>
> This is the reason our task is referred to as Neural Subset Selection. The key distinction between Neural Subset Selection and (Core) Subset Selection is that our tasks possess the ground truth of the optimal subsets. For instance, in the set anomaly detection (Figure 4&5 in [1]), several images within an image set exhibit notable differences from the others based on certain properties.
>
> In the case of (core) subset selection, there may not be an optimal subset as each item contributes some positive utility, necessitating the cardinality constraint. Such problems often employ methods that maximize a submodular function [3, 4, 5]. However, from my perspective, these are two distinct tasks. To prevent any confusion among readers, we will include a paragraph in our related work to discuss the differences between the two tasks. We would greatly appreciate any references you could provide, and kindly correct us if we have missed any.
>
> ***What is the objective function for training?***
>
> We follows the objective function from [1], which is as follows. (Apologies for the error in the first line; it should be $log p_\theta (S | V)$ intead of $log p (S | V),$ We retained the mistake due to the presentation issues on OpenReview.)
>
> $$
> argmax_\theta\  \mathbb{E}_{\mathbb{P}(V, S)} [log p (S | V)]
> $$
>
> $$
> s.t.  p_\theta (S | V) \propto  F_\theta (S ; V), \forall  S \in 2^V,
> $$
> where the constraint admits the learned set function to obey the objective defined in
> \begin{align}
>     S^* = argmax_{S \in 2^{V}} F_\theta (S; V).
> \end{align}
> Since our work mainly concentrates on the model, we offer a concise explanation in the second paragraph of the **Introduction** and encourage readers interested in the objective to refer to **Appendix D.3** for further details.
>
> **What is the initiation method of $\xi$?**
>
> Thank you for the reminder. We have included the initialization method for \xi = {s_1, s_2, \dots, s_m} in our revised draft. Specifically, we initialize it by sampling a random initialization for  $m$ embeddings $\xi \in \mathbb{R}^{m \times d_s}$ where $d_s$ is the dimension of each embedding. Specifically,
>
> \begin{equation}
>   \xi \sim \mathcal{N}(\mu, \text{diag}(\sigma)) \in \mathbb{R}^{m \times d_s},
> \end{equation}
>
> where $\mu \in \mathbb{R}^{1 \times d_s}$ and $\sigma \in \mathbb{R}^{1 \times d_s}$ are learnable parameters. This method is inspired from [6], which has been demonstrated to be effective in slot-based attention.
>
> ***Do $h(V)$ and $h(S)$ share the same $\xi$ ?***
>
> Indeed, they share the same $\xi$. Here, $[S_1, S_2, \dots, S_m]^T$ is constructed from $V$ using a partitioning method. We employ $h(S_i)$ to emphasize that the calculations of $h(S_i)$ are induced by V and its corresponding partitions, achieved by splitting the input matrix $V$ into several parts, as described in Algorithm 1 in our appendix. Throughout the calculation, any other input remains the same for $h(S_i)$ and $h(V)$. Therefore, we appreciate your suggestion to use $h_V(S)$, as it is more meaningful.
>
> ***Why we need Theorem 4.1 and what is the definition of permutation matrix?***
>
> Sets are inherently permutation invariant by construction. However, since models process sets in matrix form, a general MLP cannot guarantee the permutation invariance property. This is the reason we require Theorem 4.1. As previous works, such as [1,7], share this consensus, we do not emphasize that the input is a matrix. Furthermore, designing models to satisfy such permutation-invariant properties is an important research topic, and numerous works have been dedicated to it, such as [1, 7, 8].
>
> Furthermore, you are correct that a permutation matrix is a square binary matrix with exactly one entry of 1 in each row and each column, and all other entries being $0$, as defined in **Linear Algebra**. Here, $\pi_S \in \mathbb{R}^{n_i \times n_i}$ and $\pi_V \in \mathbb{R}^{n \times n}.$ In order to accommodate readers with a variety of backgrounds, we have incorporated your suggestions and revised the description of Theorem 4.1, emphasizing that the input is in matrix form and defining the permutation matrix $\pi.$
>
> Thank you again for your valuable time and efforts in reviewing our manuscript. We would appreciate knowing if you have any additional feedback or suggestions.
>
> ------------------------
> [1] Ou et al., "Learning Neural Set Functions under the Optimal Subset Oracle," NeurIPS, 2022.
>
> [2] Tschiatschek et al., "Differentiable Submodular Maximization," IJCAI, 2018.
>
> [3] Wei et al., "Submodularity in Data Subset Selection and Active Learning," ICML, 2015.
>
> [4] Mirzasoleiman et al., "Coresets for Data-Efficient Training of Machine Learning Models," ICML, 2020.
>
> [5] Yang et al., "Towards Sustainable Learning: Coresets for Data-Efficient Deep Learning," ICML, 2023.
>
> [6] Locatello et al., "Object-Centric Learning with Slot Attention," NeurIPS, 2020.
>
> [7] Zaheer et al., "Deep Sets," NIPS, 2017.
>
> [8] Maron et al., "On Learning Sets of Symmetric Elements," ICML, 2020.

---

> ### Author Response · Authors · 2024-08-11
>
> Dear Reviewer aVVy,
>
> We sincerely appreciate your valuable and helpful suggestions. We are wondering whether you have any more suggestions or questions after our response. Specifically, do you have any questions on our training objective or do you have any reference about Corset Selection to help use differentiate Neural Subset Selection with Corset Selection. Moreover, if you find our response satisfactory, we kindly invite you to consider the possibility of improving your rating.
>
> Sincerely,
> Authors

---

### Official Review · Reviewer_tohe · 2024-08-03

**Soundness:** 3
**Presentation:** 3
**Contribution:** 3
**Rating:** 6
**Confidence:** 4

**Summary:**

The paper introduces HORSE, a method for neural subset selection. It addresses the limitations of existing methods by introducing the concept of Identity Property and utilizing an attention-based mechanism to capture complex interactions within the input set. HORSE demonstrates superior performance on various tasks, including product recommendation and compound selection.

**Strengths:**

- The paper introduces the novel concept of the Identity Property for neural subset selection, which is a valuable contribution to the field. The attention-based approach to modeling interactions within sets is also a creative and effective solution.
- The paper demonstrates a strong theoretical foundation with clear explanations of the proposed method. The experimental evaluation is comprehensive, covering multiple datasets and tasks, providing strong evidence for the method's effectiveness.
- The paper is well-structured and easy to follow, with clear explanations of complex concepts. The authors effectively communicate the motivation, methodology, and results of the work.
- By addressing the limitations of existing methods and demonstrating superior performance on various tasks, the paper offers a significant contribution to the field of neural subset selection.

**Weaknesses:**

- While the paper includes several experiments, a more extensive evaluation with a wider range of datasets and tasks would strengthen the claims. It would be beneficial to compare the proposed method to a broader set of baselines, including recently proposed methods for set-based tasks.
- A more detailed ablation study to isolate the contributions of different components of the proposed method (e.g., attention mechanism, partitioning scheme) would provide deeper insights into the model's effectiveness.
- The paper could provide more details about the computational complexity of the proposed method and compare it to other approaches. This information would be valuable for practical applications.
- Although the paper mentions handling large-scale inputs, a more thorough evaluation on extremely large datasets would demonstrate the scalability of the proposed method.
- While the paper provides theoretical analysis of the Identity Property and permutation invariance, a deeper theoretical investigation into the properties of the attention mechanism and its impact on the model's performance could provide additional insights

**Questions:**

- How does HORSE compare to other state-of-the-art methods for subset selection, such as those based on graph neural networks or deep sets with attention mechanisms?
- Could the authors provide a more detailed ablation study to analyze the impact of different components of the proposed method, such as the attention mechanism, the partitioning strategy?
- What is the computational complexity of HORSE compared to other methods, especially for large-scale datasets? How does the choice of partitioning strategy affect computational efficiency?
- Can the authors provide further theoretical insights into the properties of the attention mechanism and its relationship to the Identity Property?
- How does HORSE handle imbalanced datasets or noisy data? What are the potential challenges and limitations of applying HORSE to real-world applications?

**Limitations:**

# Limitations already addressed
- The paper acknowledges limitations like sensitivity to hyperparameter tuning (through potential need for careful hyperparameter search) and potential overfitting (although not extensively discussed).

# Limitations NOT addressed

- The impact of noise in the data is not explicitly discussed.
- The paper mentions handling large-scale inputs but lacks thorough evaluation on extremely large datasets, leaving scalability concerns unaddressed.

---

> ### Author Rebuttal · Authors · 2024-08-07
>
> We greatly appreciate the time and effort you have invested! In response to your concerns and insightful questions, we have provided detailed clarifications and additional experimental results.
>
> ***Are there any state-of-the-art GNN-based or deep sets models that utilize attention mechanisms?***
>
> Most GNN-based methods for subset selection are engineered to select a subset from the entire training dataset for efficient model training, as seen in examples like [1, 2]. This is commonly referred to as Coreset Selection. In this context, there is no supervision for the optimal subset. Thus, these methods cannot be directly applied to our tasks.
>
> Regarding deep sets with attention mechanisms, we also use the set-transformer [3] as a baseline. To include a broader and more up-to-date set of baselines, we have added two more: Set-T (ISAB) denotes one of the variants of the Set Transformer [3], and EquiVSet-T represents the addition of EquiVSet [4] with an self-attention mechanism. Please note that EquiVSet is one of the state-of-the-art methods in the Neural Subset Selection field. Due to time constraints and for your convenience, we have only reported a portion of the results for the Product Recommendation tasks. We will include the full results in our revised version once it is completed.
>
> | | Random  | Set-T (ISAB) | EquiVSet-T | HORSE |
> | ---------- | ----------------- | ------------------------- |------------------------- | ------------------------- |
> | Toys     | 0.083           | 0.637 ± 0.018          | 0.743 ± 0.014 |             **0.774±0.009**            |
> | Gear     | 0.077           | 0.642 ± 0.009          | 0.755 ± 0.010 |              **0.832±0.013**      |
> | Carseats | 0.066           | 0.225 ± 0.011          | 0.224 ± 0.025 |              **0.248±0.022**   |
> | Bath     | 0.076           | 0.736 ± 0.009          | 0.802 ± 0.006 |         **0.876±0.010**         |
> | Health   | 0.076           | 0.701 ± 0.013          | 0.772 ± 0.009 |         **0.816±0.006**          |
> | Diaper   | 0.084           | 0.796 ± 0.007          | 0.867 ± 0.007 |         **0.880±0.008**       |
> | Bedding  | 0.079           | 0.772 ± 0.014          | 0.798 ± 0.011 |         **0.871±0.007**      |
> | Feeding  | 0.093           | 0.763 ± 0.008          | 0.852 ± 0.006 |          **0.903±0.011** |
>
> ***More detailed ablation study to analyze the impact of different components of the proposed method.***
>
> Thank you for your suggestion. There are two main choice will influence our method's performance, i.e., pooling and partitioning method. Firstly, we report various choices of pooling methods as the aggregation function to demonstrate their impacts. Due to space limitation, we kindly invited your to refer table 1 in the following comments window. These four methods have similar overall performance, with sum pooling and mean pooling being slightly better. We use mean pooling in our method across different tasks and datasets.
>
> For the partitioning strategy, we use random partitioning, which is a common choice. Indeed, the number of partitions will impact the performance of HORSE. We have presented Figure 3 to demonstrate how variations in the number of partitions and the size of each partition will impact the results. While all neural subset methods will experience a drop in performance when the number of partitions increase, HORSE exhibits significantly more **robust and superior performance**. This implies that HORSE will be more robust in large-scale settings.
>
> ***What is the computational complexity of HORSE compared to other methods.***
>
> Our framework shares a simlar framework  with our baselines, EquiVSet and INSET. When considering only the attention mechanism, our method's time complexity is O(nk), which is lower than that of self-attention-based methods such as EquiVSet-T, which has a time complexity of O(n^2). Therefore, computational complexity should not be a major concern, as HORSE is more scalable and performs better and faster than EquiVSet-T, especially on extremely large datasets.
>
> ***Provide further theoretical insights into the properties of HORSE and its relationship to the Identity Property***
>
> We provide Theorem 4.2 in our paper to illustrate that the attention mechanism we proposed adheres to the Identity Property. This suggests that HORSE can process and aggregate each subset from a set partition, resulting in the same representation as encoding the entire set at once.
>
> ***How does HORSE handle imbalanced datasets or noisy data?***
>
> Due to space limitations and for your convenience, we have included some empirical studies in the following comments section. We kindly invite you to review them. These results demonstrate that HORSE can still achieve better performance than our baselines.
>
> ***What are the potential challenges and limitations of applying HORSE to real-world applications?***
>
> We concur that in real-world applications, noisy data and distribution gaps may diminish the performance of HORSE, an issue also overlooked by other neural subset selection methods. As such, we eagerly anticipate future research that could potentially enhance the robustness of these methods. From a deployment perspective, there are additional concerns, such as communication and synchronization issues, which fall outside the scope of our paper.
>
> Thank you for your time. If you have any additional questions, we would be delighted to discuss them further.
>
> -------------------------------------------
> [1] Breustedt et al., "On the Interplay of Subset Selection and Informed Graph Neural Networks," 2023.
>
> [2] Jain et al., "Efficient Data Subset Selection to Generalize Training Across Models: Transductive and Inductive Networks," 2024.
>
> [3] Lee et al., "Set Transformer: A Framework for Attention-Based Permutation-Invariant Neural Networks," 2019.
>
> [4] Ou et al., "Learning Neural Set Functions under the Optimal Subset Oracle," 2022.

---

> ### Author Response · Authors · 2024-08-07
> **Some Empirical Studies Complementing Our Rebuttal**
>
> ***Ablation Studies on the pooling methods***
>
> Our method employs pooling methods as our aggregation function. Following the common selection in set-related works, such as [1, 2], we use mean pooling, as it typically yields better performance. In our studies, we also find that these four methods exhibit similar overall performance, with sum pooling and mean pooling being slightly superior.
>
> |   | mean pooling |  sum pooling          | min pooling          | max pooling
> | ----------- | -------------- | -------------- | -------------- | -------------- |
> | PDBBind      | **0.941 ± 0.007** | 0.939±0.010 | 0.934±0.013 | 0.935±0.012 |
> | BindingDB   | 0.742 ± 0.011 | **0.745±0.012** | 0.739±0.015 | 0.736±0.019 |
> | PDBBind-2      | 0.432 ± 0.006 | **0.435±0.008** | 0.423±0.009 | 0.425±0.012 |
> | BindingDB-2      | **0.213 ± 0.009** | 0.210±0.007 | 0.211±0.011 | 0.206±0.014 |
>
> ***How does HORSE handle imbalanced datasets or noisy data?***
>
> In neural subset selection tasks, the imbalance may arise from differences in the size of the optimal subsets and the size of the ground set. To provide empirical evidence, we conduct additional experiments that showcase INSET's consistent superiority over the baselines, even in scenarios with imbalanced ground set sizes. Specifically, we train the model on the two-moons dataset using fixed optimal subset sizes of 12 and evaluate its performance on various ground set sizes ranging from 200 to 1000.
>
> |   | 200 |  400        | 600 | 800 | 1000|
> | ----------- | -------------- | -------------- | -------------- | -------------- | -------------- |
> | EquiVSet      | 0.486 ± 0.002 | 0.453 ± 0.003 | 0.432 ± 0.002 | 0.425 ± 0.005 | 0.396 ± 0.003  |
> | INSET | 0.494 ± 0.002 | 0.478 ± 0.005|  0.447 ± 0.003 | 0.434 ± 0.002 | 0.425 ± 0.002 |
> |HORSE | **0.527 ± 0.004** | **0.503 ± 0.006** |  **0.476 ± 0.005** | **0.466 ± 0.002** | **0.453 ± 0.003** |
>
> For imbalanced datasets, we carry out additional experiments using the CelebA Dataset to underscore INSET's consistent superiority over the baseline models. Specifically, we will randomly insert one wrong image  in the optimal subset for anomaly detection tasks. From the results, the performance of all methods drops significantly, while HORSE still outperforms the others. This inspires further promising research on the topic to make neural subset selection methods more robust against noisy labels.
>
> |   | PGM |  DeepSet        | Set-T  | EquiVSet | INSET |    HORSE     |
> | ----------- | -------------- | -------------- | -------------- |  -------------- | -------------- | -------------- |
> | CelebA      | 0.371±0.027 | 0.436±0.031 | 0.417±0.028 | 0.437±0.035 |  0.465±0.037  | **0.469±0.034**  |

---

### Decision · Program_Chairs · 2024-09-25

**Decision:**

Accept (poster)

**Comment:**

**Summary of the paper:**
The paper deals with the supervised learning of deep networks for subset selection. The paper identifies expressivity issues with the recent best-performing methods, particularly, in modeling the interaction within and between the original set elements and the optimal subset elements. It then defines a desirable formal property on set representations that their proposed model, HORSE, satisfies. HORSE uses an attention mechanism to model the desired interactions and furthermore uses a random partitioning of the full set for a more efficient handling of large sets. Several experiments including, two on synthetic data, few on proteins binding, and one on a product recommendation task, are conducted exhibiting consistent improvements over the baselines.

**Summary of the reviews:**
The reviewers appreciate the importance of the problem, the relevance of “identity property” for dealing with large input sets and the simplicity of the attention mechanism in modeling the desired interactions. They further found the experiments substantiating the claim. Most reviewers found the presentation well done.

On the other hand, the reviewers asked for a broader set of benchmarks and baselines, particularly large-scale sets. They further brought up the importance of ablation study for different components and asked for a discussion on the role of partitioning. Errors and complicatedness of the notation were also raised.

**Summary of the rebuttal and discussions:**
The authors provided (and referred to) the ablation study of the various pooling functions and the different partitioning scenarios, and further discussed the role and the effect of partitioning. They further compared with more advanced baselines equipped with attention and introduced a couple of new benchmarks during the rebuttal. They acknowledge the possible improvements in the notation.

**Consolidation report:**
The AC agrees with the reviewers' positive points on the relevance and significance of the problem and the proposed approach. The AC believes the concerns are mitigated with the rebuttal regarding the benchmarks, baselines, the ablation studies and the choice of partitioning. The notation can be improved but is relatively minor and can be done in a camera-ready submission.

**Recommendation:**
The AC agrees with the unanimous recommendation by the reviewers and suggests acceptance.